# Changes in Eyelid Pressure and Dry Eye Status after Orbital Decompression in Thyroid Eye Disease

**DOI:** 10.3390/jcm10163687

**Published:** 2021-08-20

**Authors:** Yasuhiro Takahashi, Aric Vaidya, Hirohiko Kakizaki

**Affiliations:** 1Department of Oculoplastic, Orbital & Lacrimal Surgery, Aichi Medical University Hospital, Aichi 480-1195, Japan; aricvaidya1@gmail.com (A.V.); cosme_geka@yahoo.co.jp (H.K.); 2Department of Oculoplastic, Orbital & Lacrimal Surgery, Rapti Eye Hospital, Dang 22412, Nepal

**Keywords:** eyelid pressure, dry eye, meibomian gland dysfunction, superior limbic keratoconjunctivitis, thyroid eye disease, orbital decompression

## Abstract

The aim of this prospective observational study was to examine changes in eyelid pressure and dry eye status after orbital decompression in thyroid eye disease (TED). In 16 patients (29 sides), upper eyelid pressure at plateau phase and maximum pressure were measured. TED status was evaluated through the Hertel exophthalmometric value and margin reflex distance (MRD)-1 and 2. Dry eye status was quantified through corneal fluorescein staining, tear break-up time, Schirmer test I results, meibomian gland dysfunction (MGD), tear meniscus height, and superior limbic keratoconjunctivitis (SLK). Patients were classified into two groups: patients with decreased eyelid pressure (Group 1) and those with elevated pressure (Group 2). Consequently, neither the maximum upper eyelid pressure nor pressure at plateau phase significantly changed after surgery (*p* > 0.050). Some parameters about MGD improved after surgery, but the other parameters on dry eye, MGD, and SLK worsened or did not change. MRD-1 decreased more (*p* = 0.028), and the ratio of patients in whom SLK improved after surgery was larger in Group 1 (*p* = 0.030). These results indicate that upper eyelid pressure tends to decrease postoperatively in patients with a high upper eyelid position, resulting in improvement of SLK.

## 1. Introduction

The eyelid pressure is regulated by the eyelid tension and anterior eye position [1]. In thyroid eye disease (TED), an enlarged and cicatricial levator palpebrae superioris produces a taut upper eyelid as well as upper eyelid retraction [2,3]. Proptosis also stretches the upper and lower eyelids [4]. Previous experimental studies showed that tension of the eyelid in the anterior direction increases eyelid tension [5,6]. A high eyelid pressure can cause abnormal friction between the ocular surface and palpebral conjunctiva [2]. This increases concentration of inflammatory cytokines and induces local mucin deficiency in the ocular surface [2,7,8], resulting in dry eye and superior limbic keratoconjunctivitis (SLK) in TED.

Orbital decompression primarily improves proptosis and secondarily eyelid retraction [4,9]. This implies that orbital decompression may reduce eyelid pressure and subsequently improve dry eye and SLK. A previous study showed improvement of SLK after orbital decompression [10]. However, there had been no previous reports on the relationship between orbital decompression and eyelid pressure.

Here, we examined the changes in eyelid pressure and dry eye status, including meibomian gland dysfunction (MGD) and SLK, after orbital decompression in TED.

## 2. Materials and Methods

### 2.1. Study Design and Patients

This was a prospective observational study including Japanese patients with TED who underwent orbital decompression for disfiguring proptosis and approved participation in this study from January 2016 to January 2017. All patients sought treatment for disfiguring proptosis by themselves, consulted with us, and decided to undergo orbital decompression. When a patient underwent bilateral orbital decompression, both eyes/sides were included in this study because TED can present differently between the two eyes/sides in the same patient [11]. A diagnosis of TED was made based on the presence of at least one characteristic sign (eyelid fullness, eyelid retraction, proptosis, and/or restrictive strabismus) and the presence of thyroid autoimmunity [12]. All patients were euthyroid at the time of surgery and in the static phase of TED, based on 3 points or less in a 7-point scale of the clinical activity score [13]. None of the patients showed severe corneal involvement due to lagophthalmos or compressive optic neuropathy, had undergone previous eyelid or orbital surgery, orbital radiotherapy, or had any history of contact lens use or topical lubricants. All patients showed a patent lacrimal drainage system confirmed by lacrimal syringing.

### 2.2. Measurements

All measurements were performed by one of the authors (Y.T.) 1 day before and 3 months after surgery.

#### 2.2.1. Eyelid Pressure

Eyelid pressure was measured using a tactile pressure sensor (DigiTacts Single Point Sensors; Pressure Profile Systems, Inc, Los Angeles, CA) according to our previous study and the study conducted by Sakai et al. [1,14]. Since the sensor tip was not waterproof, this was covered with a disposable waterproof polyurethane cap 0.03 mm thick (Okamoto Industries, Inc., Tokyo, Japan). A new cap was used for each patient to prevent the spread of infection. The pressure sensor was connected to a personal computer, which took measurements automatically every 0.03 s.

After the instillation of topical 0.4% oxybuprocaine (Santen, Osaka, Japan), each patient was fitted with a sterile disposable soft contact lens (−0.5 diopters, Focus DAILIES; CIBA Vision, Duluth, GA) to protect the cornea. The pressure sensor was then inserted into the center of the upper conjunctival fornix. The patients were asked to keep their eyes open until the measured pressures reached a plateau phase that was maintained for 5 s (Figure 1). They were then directed to close their eyes forcefully three times. The mean maximum pressure during forceful eyelid closure was calculated (Figure 1). The results of lower eyelid pressure were not used for statistical comparison because the lower crus of the lateral canthal supporting structure was temporarily disinserted during deep lateral orbital wall decompression, as mentioned below.

#### 2.2.2. TED Condition

Hertel exophthalmometry values and margin reflex distance (MRD)-1 and -2 were assessed. In Hertel exophthalmometry measurements, the distance from the corneal apex to a plane, defined by the deepest point on the lateral orbital rim, was measured. A base value, the distance between the two footplates, was documented at the first measurement and reproduced at the follow-up examinations. MRD-1 and MRD-2 were measured as the distance from the upper (MRD-1) or lower (MRD-2) eyelid margin to the corneal light reflex in the primary eye position. The patient was set in the sitting position with brow fixation and was requested to look at a light source (a pen torch), then the distances were recorded using a millimeter ruler.

#### 2.2.3. General Assessment of Dry Eye

The area (A) and density (D) classification of corneal fluorescein staining, tear break-up time (TBUT), and Schirmer test I were used to assess the severity of dry eye. The AD classification was graded using the scale reported by Miyata et al. (Table 1) [15]. The TBUT was determined by fluorescein staining of the ocular surface. The time just after eye-opening to the first appearance of a dry spot on the cornea was measured. Schirmer test I was performed without anesthesia as follows: a Schirmer test strip was placed in the lower conjunctival sac without touching the cornea, and the length of the wet portion after 5 min was measured.

#### 2.2.4. TMH

TMH was measured on the sagittal plane through the center of the upper and lower eyelids using optical coherence tomography (OCT) (RS-3000, NIDEK CO., LTD, Aichi, Japan) [14,16,17]. The dedicated attachment was used to observe the anterior segment of the eyes.

#### 2.2.5. MGD

MGD was assessed through 4 criteria: the presence or absence of eyelid abnormalities, the position of Marx line, the quality and ease of meibum expression, and loss of the meibomian glands, as based on our previous studies [14,18,19].

The presence or absence of eyelid abnormalities (irregular lid margin, vascular engorgement, plugged meibomian orifices) was expressed using a binary system (a dummy variable; 0 = absence, 1 = presence), and total sum scores (MGD score) were calculated in each upper and lower eyelid for statistical analysis. The maximum MGD score was 6 points. The position of the Marx line was determined by fluorescein staining of the ocular surface. The resulting stained lines along the eyelids were examined by slit-lamp biomicroscopy after several blinks. The grading scale reported by Yamaguchi et al. was used (Table 1) [20]. The eyelid was further divided into 3 segments (the outer third, middle third, and inner third), and the position of the Marx line was evaluated in each; the maximum score for each eyelid was 9. The grading of quality and ease of meibum expression that we previously defined was used (Table 1) [14,18,19]. The loss of meibomian glands in the eyelids was evaluated using a mobile pen-shaped meibography (Meibom Pen; Japan Focus Co., Ltd.; Tokyo; Japan). The grading scale previously reported by Arita et al. was used (Table 1) [21].

#### 2.2.6. SLK

Patients with at least 2 of the following criteria were diagnosed with SLK: blood vessel dilation in the superior bulbar conjunctiva, papillary inflammation of the upper tarsal conjunctiva, punctate fluorescein staining of the superior conjunctiva and the upper cornea, filaments in the upper cornea, epithelial thickening of the superior bulbar conjunctiva, and redundancy of the superior bulbar conjunctiva [2].

### 2.3. Surgical Procedures

All surgeries were performed under general anesthesia with the aid of binocular loupes (high-resolution prismatic HRP × 2.5, 340 mm/13 inches; Heine, Herrsching, Germany) by one of the authors (Y.T.). The details are described in our previous reports [22,23].

#### 2.3.1. Deep Lateral Orbital Wall Decompression

A 20 mm skin incision along the lateral canthal rhytide with a lateral cantholysis of the lower crus was made. After incising the periosteum, the lateral orbital wall and lateral half of the orbital roof were exposed. Bone removal was performed using an ultrasonic bone aspirator (Sonopet UST-2000R; Stryker Japan, Tokyo, Japan) up to the cortical bone of the posterior, superior, and lateral borders of the greater wing of the sphenoid bone. After incising the periosteum of the lateral wall, orbital fat was removed from the inferolateral quadrant of the orbit, if necessary.

#### 2.3.2. Medial Orbital Wall Decompression

After an incision on the lacrimal caruncle, blunt dissection was performed towards the medial orbital wall 5 mm posterior to the posterior lacrimal crest. After an incision on the periosteum just below the origin of Horner’s muscle, the periosteum was reflected from the medial orbital wall. The lamina papyracea was infractured, and the bone was removed. Bone removal reached anteriorly to the point just below the origin of Horner’s muscle, posteriorly to the posterior ethmoidal foramen, superiorly to the frontoethmoidal suture, and inferiorly to the junction between the medial orbital wall and orbital floor to preserve the inferomedial orbital strut. A periosteal flap with a posterior hinge was made along the medial rectus muscle.

### 2.4. Statistical Analyses

Patient age and measurement results are expressed as the mean value ± standard deviation. Patient sex was expressed using a binary system (a dummy variable; 0 = male, 1 = female). Changes in measurement values were calculated by subtraction of postoperative results from preoperative values. The measurement results were compared before and after surgery using paired t-test or Wilcoxon signed-rank test. Patients were classified into 2 groups: patients in whom both the maximum eyelid pressure and eyelid pressure at plateau phase decreased (Group 1) and those in whom either or both the pressure elevated (Group 2). The measurement results were compared between the groups using the Mann–Whitney U test. Univariate and following multivariate linear regression analyses with stepwise variable selection were performed to identify factors influencing changes in eyelid pressure. The predictive variables investigated were patient age; sex; preoperative MRD-1, MRD-2, and Hertel exophthalmometric value; and postoperative changes in MRD-1, MRD-2, and Hertel exophthalmometric value. All statistical analyses were performed using SPSS ver. 26 software (IBM Japan, Tokyo, Japan). A *p*-value of <0.050 was considered statistically significant.

## 3. Results

Sixty-two sides from 34 patients underwent orbital decompression during the study period, but 18 patients declined participation in this study. This study included 29 sides from 16 patients (3 males and 13 females; 15 right and 14 left; mean age, 38.5 ± 10.1 years). Deep lateral orbital wall decompression and balanced orbital decompression were performed on 21 and 8 sides, respectively.

The measurement results are shown in Table 2. Neither the maximum upper eyelid pressure nor upper eyelid pressure at the plateau phase significantly changed after orbital decompression (*p* > 0.050). As for the TED condition, Hertel exophthalmometric results and MRD-2 significantly decreased postoperatively (*p* < 0.050), whereas MRD-1 did not significantly change after surgery (*p* = 0.832). In regard to dry eye status, D score (*p* = 0.026), TBUT (*p* = 0.068), and the result of Schirmer test (*p* < 0.001) tended to deteriorate after surgery, although A score did not change postoperatively (*p* = 0.932). Concerning MGD status, the Marx line score in the upper eyelid (*p* = 0.080), MGD score (*p* = 0.008), and meibography score in the upper eyelid (*p* = 0.008) tended to improve after surgery, while the Marx line score in the lower eyelid tended to get worse postoperatively (*p* = 0.078). The other measurement items did not change after surgery (*p* > 0.050). The ratio of patients with and without SLK did not change after orbital decompression (*p* = 1.000).

The results of the comparison of postoperative changes in measurements between the groups are shown in Table 3. Both the maximum upper eyelid pressure and upper eyelid pressure at the plateau phase decreased on 13 sides (Group 1, 44.8%). Hertel exophthalmometric value tended to be reduced more in Group 2 (*p* = 0.068), while MRD-1 decreased more in Group 1 (*p* = 0.028). The ratio of patients in whom SLK improved after surgery was larger in Group 1 (*p* = 0.030). The other measurement items were not significantly different between the groups (*p* > 0.050), except the eyelid pressure.

The results of the univariate and multivariate linear regression analyses are shown in Table 4. The univariate analysis showed that preoperative MRD-1 (*p* = 0.016) and changes in MRD-1 (*p* = 0.001) were correlated with changes in the maximum upper eyelid pressure. In the multivariate linear regression analysis, only the change in MRD-1 was the predictive factor of changes in maximum upper eyelid pressure (*r* = 0.586; adjusted *r*^2^ = 0.343; *p* = 0.001). In addition, in the univariate linear regression analysis, only the change in MRD-1 was the predictive factor of changes in the upper eyelid pressure at the plateau phase (*r* = 0.540; adjusted *r*^2^ = 0.291; *p* = 0.003). The amount of decrease in MRD-1 was positively correlated with the amount of decrease in both the maximum eyelid pressure and eyelid pressure at the plateau phase. There was no multi-collinearity among independent variables (range of variance inflation factor, 1.014 to 1.508).

## 4. Discussion

The present study examined changes in the upper eyelid pressure and status of dry eye after orbital decompression. We had suspected that proptosis is a possible factor that regulates eyelid pressure and dry eye. However, although proptosis and lower eyelid retraction significantly improved postoperatively, neither the maximum upper eyelid pressure nor the upper eyelid pressure at the plateau phase significantly reduced. Similarly, proptosis reduction was less in Group 1 (patients with decreased eyelid pressure). Furthermore, although some parameters about MGD improved after surgery, the other parameters on dry eye, MGD, and SLK worsened or did not change. These results were contradictory to our speculation.

On the contrary, Group 1 showed decreased upper eyelid pressure and MRD-1 and a higher ratio of patients with improvement of SLK after surgery compared with Group 2. Furthermore, the amount of decrease in MRD-1 was positively correlated with the amount of decrease in both the maximum eyelid pressure and eyelid pressure at the plateau phase. We additionally compared the ratio of patients with upper eyelid retraction between the groups. Although the difference did not reach statistical significance, Group 1 included more patients with upper eyelid retraction (*p* = 0.143, Fisher’s exact test). These results indicate that upper eyelid pressure and MRD-1 tend to decrease after orbital decompression in patients with a longer MRD-1, resulting in a decrease in abnormal friction between the upper bulbar conjunctiva and palpebral conjunctiva in the upper eyelid and leading to SLK improvement.

The Marx line, MGD, and meibography scores improved after orbital decompression in this study. Previous studies showed that the relationship between MGD and TED, incomplete blinking due to proptosis and eyelid retraction, and pathological changes in the levator palpebrae superioris muscle are possible etiologic factors of MGD in TED [24,25]. Orbital decompression improved proptosis and eyelid retraction, which may contribute to the improvement of MGD in this study.

On the contrary, the D score, TBUT, and the result of the Schirmer test worsened after orbital decompression. One possible cause is an improvement of ocular surface exposure due to reduced proptosis and palpebral fissure height. This reduced the reflex tear secretion, resulting in deterioration of the pre-existing dry eye.

None of the patients underwent orbital radiotherapy. This is used for active TED to reduce inflammation and to prevent recurrence of active TED [26,27]. However, as chronic dry eye occasionally occurs after radiotherapy [28], MGD and SLK may be refractory after orbital decompression in such patients. Future studies are necessary for evaluation of post-decompression dry eye condition in patients with a history of orbital radiotherapy.

In this study, the TMH was measured in a non-invasive and reliable way with the help of OCT. Furthermore, the corneal morphological changes and adhesive properties of the ocular surface can also be evaluated using OCT [29,30]. Especially, ocular surface adhesiveness decreases in patients with aqueous-deficient dry eye and those with MGD [29], both of which are frequently shown in patients with TED. Poor ocular surface adhesiveness may therefore be demonstrated in TED. Future studies will be helpful to clarify the changes in the ocular surface condition after orbital decompression in TED.

Our study was limited by several factors. First, we mixed patients who underwent deep lateral orbital wall decompression and balanced orbital decompression. However, as changes in the maximum eyelid pressure (*p* = 0.649; Mann–Whitney U test) and eyelid pressure at plateau phase (*p* = 0.153; Mann–Whitney U test) were not significantly different between patients who underwent deep lateral orbital wall decompression and balanced decompression, we believe that the influence of inclusion of mixed patients was not large. Second, the inclusion of only Japanese patients was another limitation as the periocular anatomy has racial differences [31]. Third, the number of patients included in this study was relatively low. Fourth, the measurement results of lower eyelid pressure were excluded from this study. Nowadays, we use the eyelid crease approach for deep lateral orbital wall decompression in patients with eyelid crease, but previously we used to cut the lower crus of the lateral canthal supporting structure during deep lateral orbital wall decompression. Fifth, it would be better to include a control group of patients with TED who had not undergone orbital decompression, with similar measurement fashion.

## 5. Conclusions

In conclusion, we examined the changes in upper eyelid pressure and dry eye status after orbital decompression. In total, the upper eyelid pressure did not change significantly. Although the MGD improved postoperatively, we did not obtain clear-cut results of improvement or deterioration of dry eye after surgery. On the other hand, upper eyelid pressure tends to decrease postoperatively in patients with a high upper eyelid position, resulting in improvement of SLK.

## Figures and Tables

**Figure 1 jcm-10-03687-f001:**
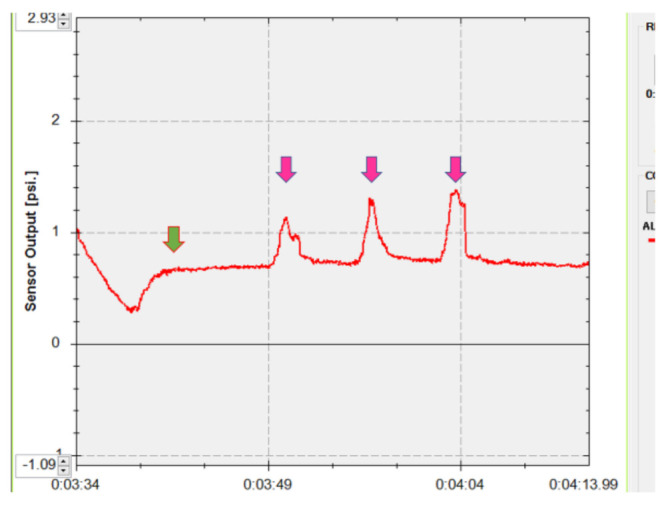
Measurement of the upper eyelid pressure. The patient kept his/her eyes open until the measured pressure became a plateau, which was continued for 5 s (green arrow). Then, the patient closed his/her eyes forcefully three times (pink arrows).

**Table 1 jcm-10-03687-t001:** The grading systems of measurement parameters.

Measurement Parameters	Grading	Findings
A	0	No punctate staining
	1	The staining involving less than one-third of the cornea
	2	The staining involving one-third to two-thirds of the cornea
	3	The staining involving more than two-thirds of the cornea
D	0	No punctate staining
	1	Sparse density
	2	Moderate density
	3	High density and overlapped lesions
Marx line	0	The line runs entirely along the conjunctival side of the meibomian gland orifices
	1	Parts of the line touch the meibomian orifices
	2	The line runs through the meibomian orifices
	3	The line runs along the eyelid margin side of the meibomian orifices
Meibum expression	0	Easy expression of clear meibum with mild eyelid compression
	1	Cloudy meibum expression with mild compression
	2	Cloudy meibum expression with moderate compression
	3	Toothpaste-like meibum expression with more than moderate compression
	4	No expression even with hard compression
Meibography score	0	No loss of meibomian glands
	1	Area loss less than one-third of the total meibomian gland area
	2	Area loss between one-third and two-thirds of total meibomian gland area
	3	Area loss more than two-thirds of total meibomian gland area

**Table 2 jcm-10-03687-t002:** The results of measurements and statistical comparison before and after orbital decompression.

	Preoperative	Postoperative	*p* Value
Maximum upper eyelid pressure (psi)	1.26 ± 0.37	1.26 ± 0.31	0.937
Upper eyelid pressure at plateau phase (psi)	0.94 ± 0.17	0.97 ± 0.16	0.405
TED condition			
Hertel exophthalmometric value (mm)	22.0 ± 2.4	17.3 ± 2.6	<0.001
MRD-1 (mm)	4.6 ± 1.8	4.6 ± 1.5	0.832
MRD-2 (mm)	6.6 ± 1.4	5.9 ± 1.0	0.001
AD score			
A	1.0 ± 1.0	1.0 ± 0.8	0.932
D	1.2 ± 1.2	1.8 ± 1.4	0.026
TBUT (s)	1.6 ± 1.2	0.9 ± 1.3	0.068
Schirmer test (mm)	20.1 ± 10.7	9.2 ± 5.6	<0.001
Marx line			
Upper eyelid	4.6 ± 1.9	4.3 ± 2.5	0.080
Lower eyelid	4.8 ± 2.0	5.4 ± 1.8	0.078
MGD score	1.6 ± 1.3	1.2 ± 1.2	0.008
Meibum expression			
Upper eyelid	1.3 ± 1.3	1.2 ± 1.4	0.638
Lower eyelid	1.2 ± 1.2	1.0 ± 1.0	0.276
Meibography			
Upper eyelid	1.1 ± 0.8	0.9 ± 0.7	0.008
Lower eyelid	0.5 ± 0.5	0.4 ± 0.6	0.157
TMH (μm)			
Upper eyelid	278.5 ± 77.7	274.5 ± 70.5	0.843
Lower eyelid	356.9 ± 106.5	361.4 ± 273.3	0.931
SLK (presence/absence)	11/18	10/19	1.000

TED, thyroid eye disease; MRD, margin reflex distance; TBUT, tear break-up time; MGD, meibomian gland dysfunction; TMH, tear meniscus height; SLK, superior limbic keratoconjunctivitis.

**Table 3 jcm-10-03687-t003:** Comparison of measurement results between the groups.

	Group 1: Decreased Pressure (*n* = 13)		Group 2: Elevated Pressure (*n* = 16)		*p* Value: Postoperative Changes, Group 1 vs. 2
	Preoperative	Postoperative	Preoperative	Postoperative	
Maximum upper eyelid pressure (psi)	1.43 ± 0.44	1.25 ± 0.42	1.12 ± 0.24	1.27 ± 0.20	<0.001
Upper eyelid pressure at plateau phase (psi)	1.04 ± 0.15	0.93 ± 0.20	0.86 ± 0.15	1.00 ± 0.12	<0.001
TED condition					
Hertel exophthalmometric value (mm)	20.7 ± 1.1	17.0 ± 2.1	23.0 ± 2.7	17.6 ± 3.0	0.068
MRD-1 (mm)	5.0 ± 1.8	4.6 ± 1.5	4.2 ± 1.7	4.5 ± 1.5	0.028
MRD-2 (mm)	6.0 ± 0.8	5.3 ± 0.3	7.2 ± 1.6	6.3 ± 1.2	0.714
AD score					
A	1.2 ± 0.8	0.9 ± 0.9	0.8 ± 1.0	1.1 ± 0.9	0.232
D	1.8 ± 1.2	1.9 ± 1.4	0.8 ± 1.0	1.8 ± 1.3	0.132
TBUT (s)	1.4 ± 1.4	1.0 ± 1.2	1.7 ± 1.1	0.9 ± 1.5	0.288
Schirmer test (mm)	19.6 ± 9.5	9.5 ± 5.7	20.4 ± 11.8	8.9 ± 5.7	0.746
Marx line					
Upper eyelid	5.3 ± 1.3	4.8 ± 2.2	4.0 ± 2.2	3.9 ± 2.8	0.983
Lower eyelid	5.4 ± 1.5	5.5 ± 1.8	4.4 ± 2.2	5.3 ± 1.9	0.398
MGD score	1.8 ± 1.2	1.5 ± 1.3	1.4 ± 1.4	1.0 ± 1.2	0.846
Meibum expression					
Upper eyelid	1.8 ± 1.6	1.2 ± 1.3	0.9 ± 1.0	1.3 ± 1.4	0.249
Lower eyelid	1.8 ± 1.5	1.1 ± 1.0	0.8 ± 0.7	0.9 ± 1.0	0.121
Meibography					
Upper eyelid	1.2 ± 0.8	0.8 ± 0.6	1.0 ± 0.8	0.6 ± 0.5	0.249
Lower eyelid	0.5 ± 0.5	0.5 ± 0.7	0.9 ± 0.8	0.3 ± 0.4	0.121
TMH (μm)					
Upper eyelid	280.5 ± 94.9	246.6 ± 41.1	276.9 ± 63.8	346.3 ± 127.7	0.423
Lower eyelid	370.0 ± 75.8	292.2 ± 83.9	297.2 ± 81.9	417.8 ± 355.3	0.215
SLK (improvement/absence-absence/deterioration or no improvement)	4/4/5		0/11/5		0.030

TED, thyroid eye disease; MRD, margin reflex distance; TBUT, tear break-up time; MGD, meibomian gland dysfunction; TMH, tear meniscus height; SLK, superior limbic keratoconjunctivitis.

**Table 4 jcm-10-03687-t004:** Results of univariate and multivariate linear regression analyses.

Dependent Variable: Maximum Eyelid Pressure	Univariate		Multivariate Stepwise	
	Crude Coefficient (95% CI)	*p* Value	Crude Coefficient (95% CI)	*p* Value
Age	0.005 (−0.006 to 0.015)	0.364		
Sex	0.142 (−0.113 to 0.397)	0.262		
Preoperative MRD-1	0.063 (0.013 to 0.114)	0.016	-	-
Changes in MRD-1	0.172 (0.078 to 0.266)	0.001	0.172 (0.078 to 0.266)	0.001
Preoperative MRD-2	−0.024 (−0.094 to 0.045)	0.476		
Changes in MRD-2	0.019 (0.077 to 0.115)	0.691		
Preoperative Hertel exophthalmometric value	−0.011 (−0.052 to 0.030)	0.573		
Changes in Hertel exophthalmometric value	−0.029 (−0.077 to 0.020)	0.238		
**Dependent Variable: Eyelid Pressure at Plateau Phase**	**Univariate**		**Multivariate Stepwise**	
	**Crude Coefficient (95% CI)**	***p* Value**	**Crude Coefficient (95% CI)**	***p* Value**
Age	0.003 (−0.005 to 0.012)	0.448		
Sex	0.024 (−0.196 to 0.245)	0.227		
Preoperative MRD-1	0.026 (−0.020 to 0.073)	0.254		
Changes in MRD-1	0.134 (0.052 to 0.217)	0.003		
Preoperative MRD-2	−0.045 (−0.101 to 0.012)	0.118		
Changes in MRD-2	−0.027 (−0.108 to 0.054)	0.568		
Preoperative Hertel exophthalmometric value	−0.020 (−0.054 to 0.013)	0.226		
Changes in Hertel exophthalmometric value	−0.035 (−0.075 to 0.005)	0.080		

MRD, margin reflex distance; CI, confidence interval.

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
