# Peer review of "Changes in Eyelid Pressure and Dry Eye Status after Orbital Decompression in Thyroid Eye Disease"

_jcm, 2021, doi:10.3390/jcm10163687_

Round 1
Reviewer 1 Report
Dear Author(s),
Thanks for your submission on the Journal of Clinical Medicine. I appreciated how the work is presented and the novelty of the features described in TED. However, the inference presented are quite weak due the relative low number of patients analysed and the mixed type of surgery.
Author Response
Reviewer #1
Thanks for your submission on the Journal of Clinical Medicine. I appreciated how the work is presented and the novelty of the features described in TED. However, the inference presented are quite weak due the relative low number of patients analysed and the mixed type of surgery.
Reply: Thank you very much for your comment.
As the reviewer indicated, inclusion of a relatively low number of patients and that of mixed patients who underwent deep lateral orbital wall decompression and balanced decompression were limitations of our study.
We performed orbital decompression for disfiguring proptosis in 34 patients (62 sides) during the study period, but 18 patients (33 sides) declined participation in this study. We added this information in page 5, “Results” section, lines 180-181, and added this limitation in page 8, “Discussion” section, 6th paragraph, lines 269-270.
We additionally compared the measurement results of the eyelid pressure between patients who underwent deep lateral orbital wall decompression and balanced decompression, but changes in the maximum eyelid pressure and eyelid pressure at plateau phase were not significantly different between these patient groups (both, P > 0.050; Mann-Whitney U test). We, therefore, believe that the influence of inclusion of mixed patients was not large. We added these contents in page 8, “Discussion” section, 6th paragraph, lines 263-267.

Reviewer 2 Report
Introduction:
- The authors provide limited data to support their hypothesis that orbital decompression may reduce eyelid pressure and improve dry eye. More supporting evidence is necessary to support the proposed hypothesis.
Methods/Study design:
It is not clear how the decision to perform orbital decompression among recruited patients was determined. Did subjects fail conservative management of dry eye disease? Was there evidence of impending compressive optic neuropathy? Please elaborate.
Was severity of TED considered in addition to disease activity?
The study would benefit from having a control group of patients with TED who had not undergone orbital decompression with similar metrics measured after 3 months.
Statistics:
- The statistical tests do not control for potentially confounding factors including age or gender.
- The statistical tests do not control for "side," treating it as an independent variable without considering 2 sides being derived from the same patient. This artificially doubles the sample size and questions the reliability of the results.
Discussion
- It is not clear whether orbital decompression improved dry or not since some of the scores improved and others did not. Please clarify.
Author Response
Reviewer #2:
Comment#1: Introduction:
The authors provide limited data to support their hypothesis that orbital decompression may reduce eyelid pressure and improve dry eye. More supporting evidence is necessary to support the proposed hypothesis.
Reply: Thank you very much for your suggestion.
We added the following sentences: “previous experimental studies showed that tension of the eyelid in the anterior direction increases eyelid tension [5,6]”; and “this increases concentration of inflammatory cytokines and induces local mucin deficiency in the ocular surface [2,7,8], resulting in dry eye and superior limbic keratoconjunctivitis (SLK) in TED” in page 1, “Introduction” section, 1st paragraph. We hope that these additional descriptions will lead to better understanding of the grounds upon which we had built up the hypothesis.
In addition, we added the following sentence: a previous study showed improvement of SLK after orbital decompression [10] in page 1, “Introduction” section, 2nd paragraph, lines 37-38.
Comment #2: Methods/Study design:
It is not clear how the decision to perform orbital decompression among recruited patients was determined. Did subjects fail conservative management of dry eye disease? Was there evidence of impending compressive optic neuropathy? Please elaborate.
Reply: Thank you very much for your indication.
We asked all patients who underwent orbital decompression for disfiguring proptosis from January 2016 to January 2017 to participate in our study. Thirty-four patients underwent orbital decompression during the study period, but 18 patients declined participation in this study. We, therefore, included 16 patients in this study. We stated these contents in page 2, 1st paragraph, lines 44-46, and page 5, “Results” section, 1st paragraph, lines 180-181.
None of the patients showed compressive optic neuropathy or used topical lubricants. We added this information in page 2, 1st paragraph, lines 52-55.
Comment #3: Was severity of TED considered in addition to disease activity?
Reply: As the reply to your comment #2, we did not include patients with compressive optic neuropathy, as well as severe corneal involvement due to lagophthalmos (so-called grade 5 and 6 of NOSPECS classification) (page 2, 1st paragraph, lines 52-54).
Comment #4: The study would benefit from having a control group of patients with TED who had not undergone orbital decompression with similar metrics measured after 3 months.
Reply: We agreed with your comment. We added this limitation of our study in page 9, “Discussion” section, 1st paragraph, lines 274-275.
Comment #5: Statistics:
The statistical tests do not control for potentially confounding factors including age or gender.
Reply: Thank you very much for your indication.
We additionally performed univariate and following multivariate linear regression analyses with stepwise variable selection to identify factors influencing changes in eyelid pressure. The predictive variables investigated were patient age, sex, preoperative MRD-1, MRD-2, and Hertel exophthalmometric value, and postoperative changes in MRD-1, MRD-2, and Hertel exophthalmometric value. Consequently, only the change in MRD-1 was the predictive factor of change in upper eyelid pressure. Furthermore, there was no multi-collinearity among independent variables. Therefore, age and sex were not potentially confounding factors. We added these contents in page 5, “Statistical Analyses” section, lines 172-176; page 7, 1st paragraph; and Table 4.
Comment #6: The statistical tests do not control for "side," treating it as an independent variable without considering 2 sides being derived from the same patient. This artificially doubles the sample size and questions the reliability of the results.
Reply: Thank you very much for your indication.
We agree with the reviewer’s comment. But we included 2 sides from the same patient because TED can present differently between the two eyes/sides in the same patient. We added this content in page 2, 1st paragraph, lines 46-48. We hope that this will be acceptable for the reviewer.
Comment #7: Discussion
It is not clear whether orbital decompression improved dry or not since some of the scores improved and others did not. Please clarify.
Reply: Thank you very much for your indication. But we did not obtain clear-cut results of improvement or deterioration of dry eye after surgery. We stated it in page 9, “Conclusion” section, lines 279-280.

Reviewer 3 Report
The authors present a papaer about "Changes in Eyelid Pressure and Dry Eye Status after Orbital Decompression in Thyroid Eye Disease".
The topic is absolutely interesting and and the authors have shown an adequate knowledge of the topic.
The design is sufficiently described in detail and the results section is clearly presented.
I would suggest to to into deeper details in the discussion section beacuse sometimes radiotherapy may be used for Thyroid Eye Disease as well reported in this recent review entitled "Radiotherapy for benign disorders: current use in clinical practice" doi 10.26355/eurrev_202105_25824.
The conclusions are balanced and coorectly define the resultws of the study.
Author Response
Reviewer #3:
Comment #1: The authors present a papaer about "Changes in Eyelid Pressure and Dry Eye Status after Orbital Decompression in Thyroid Eye Disease".
The topic is absolutely interesting and the authors have shown an adequate knowledge of the topic.
The design is sufficiently described in detail and the results section is clearly presented.
I would suggest to to into deeper details in the discussion section because sometimes radiotherapy may be used for Thyroid Eye Disease as well reported in this recent review entitled "Radiotherapy for benign disorders: current use in clinical practice" doi 10.26355/eurrev_202105_25824.
The conclusions are balanced and correctly define the results of the study.
Reply: Thank you very much for your suggestion.
We discussed about radiotherapy, TED, and dry eye in page 8, “Discussion” section, 5th paragraph, and cited the above-mentioned article as Reference #26.
